# Household food security access and dietary diversity amidst COVID-19 pandemic in rural Nepal; an evidence from rapid assessment

Dirghayu K. C.[1]*, Namuna Shrestha[1], Rachana Shrestha[2,3], Dev Ram Sunuwar[4], Anil Poudyal[1]

1 Public Health Promotion and Development Organization, Kathmandu, Nepal, 2 Public Health and Environment Research Center, (PERC), Kathmandu, Nepal, 3 Knowledge To Action (K2A), Kathmandu, Nepal, 4 Department of Nutrition and Dietetics, Nepal Armed Police Force Hospital, Kathmandu, Nepal

* dirghayu.kc01@gmail.com

## Abstract

### Background

The COVID-19 pandemic led to surging concerns about food insecurity status throughout the world. In response to global and national concerns on food and nutrition security, this study aimed to examine the prevalence and determining factors of household food insecurity and dietary diversity among people from selected rural municipalities of Lalitpur district, Nepal.

### Methods

A community-based cross-sectional study was conducted among 432 households. Pre-tested structured questionnaires were used to collect socio-demographic characteristics of the participants, household income; influence of COVID-19 on their income and livelihood, household's access to food and dietary diversity. Food insecurity was measured using the Household Food Insecurity Access Scale (HFIAS) and the Household Dietary Diversity Score (HDDS). A multivariable analysis was done using binary logistic regression model following a bivariate analysis to assess the association between the dependent and independent variables.

### Results

More than one-third (36%) of the households reported some form of food insecurity. The overall mean score for Household Dietary Diversity Score (HDDS) was 6.0 (±1.1). Multivariable logistic regression analysis showed that participants aged 41–64 years (aOR = 0.35, 95% CI: 0.21–0.59), those over 64 years (aOR = 0.22, 95% CI: 0.07–0.66), as well as those in service occupation (aOR = 0.22, 95% CI: 0.06–0.77) were associated with lower odds of HFIAS. Conversely, Participants belonging to a disadvantaged ethnic group (aOR = 2.73, 95% CI: 1.23–6.07), having no education (aOR = 3.70, 95% CI: 1.16–11.71) or primary education (aOR = 3.67, 95% CI: 1.23–9.89), and those suffering from chronic illness (aOR = 3.12, 95% CI: 1.53–6.35) were associated with higher odds of HFIAS. As for HDDS,

**Data Availability Statement:** All the relevant data are within the paper.

**Funding:** This research was funded by Nepal Health Research Council. However, the funding

organization had no role in study design, data collection, and analysis, decision to publish or preparation of this manuscript.

**Competing interests:** The authors have declared that no competing interest exist.

participants aged 41–64 years (aOR = 0.48, 95% CI: 0.28–0.83) were associated with lower odds of HDDS, while participants having no education (aOR = 10.05, 95% CI: 4.05–24.01) were associated with significantly higher odds of HDDS.

## Conclusion

Owing to the pandemic, our study showed a substantial prevalence of food insecurity among diverse community residing in rural outskirts of Kathmandu Valley, particularly among disadvantaged ethnic group and people with lower level of education. Interventions targeting these particular groups may help in improving HFIAS and HDDS among them during emergencies.

## Background

The World Health Organization (WHO) declared COVID-19 a Public Health Emergency of International Concern (PHEIC) on 30th January 2020.The global pandemic, which began on March 11[th], shut down the world immediately [1]. As suggested by the Food Agriculture Organization (FAO) of the United Nations, food insecurity occurs when there is an abrupt disruption in the availability, access, utilization, and stability of food [2]. The continuation of lockdown as the pandemic impeded all the pillars of food security led to incensed food insecurity globally together with various adverse public health consequences [3, 4]. Worldwide, several studies have been conducted targeting to evaluate the impact of the COVID-19 pandemic on food insecurity [5–8]. Evermore, evidence from a systematic review suggests that the COVID-19 pandemic had detrimental effect on food insecurity and diet quality [9, 10]. According to the recent World Food Program (WFP) estimates in 2020, over 280 million people worldwide are at a risk of acute food insecurity [2] along with 97 million suffering from chronic food insecurity indicating an alarming rate in comparison to pre-pandemic years [11].

Prior to the COVID-19 pandemic, 690 million individuals worldwide consumed fewer calories than required, which undoubtedly is a public health concern of utmost importance in terms of food consumption, dietary diversity and relevant food insecurity [12]. According to Nepal Demographic Health Survey (NDHS) 2016, in Nepal, 48% households were food-secure whereas, 20% of households were subjected to mild food-insecurity, 22% faced moderate food-insecurity, and 10% suffered severely [13]. Nepal being a least developed nation has always struggled to produce an adequate supply of food for its citizens [14] and has scuffled with its existing downfall of nutritional scenario. The addition of the unprecedented threat of the surging pandemic, has accorded nutrition and food security as a top priority by the Government of Nepal [15]. The adverse health consequences of the COVID-19 together with the measures taken by the government to stem it's transmission including restrictions on movement and operation of business and markets have resulted in social and economic meltdown further exacerbating food security issues across the country [16, 17]. The pandemic crisis has affected the livelihoods of Nepalese households, with 1 out of 10 households reporting a loss of livelihood whereas reduction in income was reported by 3 out of 10 households jeopardizing the economic scenario of not only an individual but also the whole nation [18].

As defined by FAO, "minimum dietary diversity" for the household as the consumption of food from at least four food groups of the 12 specified food groups through 24 hours period preceding the survey day [19]. Additionally, household food security and dietary diversity are linked together considering dietary diversity as another key indicator of a household's ability

to provide enough as well as diverse food to achieve good nutrition. This in turn serves as a measure of the nutritional quality of the diet to ensure adequate consumption of essential nutrients. Household's Dietary Diversity Scores (HDDS) are considered a proxy indicator to measure the economic ability of a household in accessing diverse food groups in a recommended period and an overall indicator of food security [19]. Dietary diversity and adequacy were found to have been significantly influenced by multiple socio-cultural as well as economic beliefs and taboos existing in Nepalese society [20]. Even more existing evidence suggests that a large proportion of Nepalese households also faced food insecurity and poor dietary diversity [16].

Based on a recent report from World Bank, Nepal's rural population accounts for almost four-fifths of the total population [21]. According to NDHS data 2016, nearly half (48%) of Nepal's households are food secure whereby urban households are more likely to be food secure (58%) compared to rural households (39%) [13]. This is triggered by the fact that most agricultural activities performed in rural areas lack agricultural innovation, inadequate market access and are located in remote areas with less productivity [22]. These figures are likely to increase with the continuance of disruption in agricultural production, supply, and labor market following measures taken to combat the pandemic crisis. Despite the government's palliative approach at multiple levels, a severe impact on food security among the Nepalese population was observed sequential of the pandemic [18]. However, there still exist inadequacies of evidence that can highlight the impact of COVID-19 on food security and dietary diversity in such rural context as the majority of the population resides there within.

This led to an assessment of the influence of the COVID-19 pandemic on household food security and their dietary diversity among rural communities of Nepal. This study is expected to offer critical evidence on different factors that determine the household food insecurity as well as their dietary diversity during COVID-19 pandemic or other emergencies of such nature and may support the government in effectively planning and designing policies and strategies for combating household food insecurity during imminent emergencies of this nature.

## Methodology

### Study design, study setting, and sample size

A community-based cross-sectional study was conducted between 3rd January and 3rd March, 2021among people residing in rural municipalities of Lalitpur district, Nepal. Lalitpur district comprises of one metropolitan city, two urban municipalities, and three rural municipalities [23]. The COVID-19 crisis triggered multi-dimensional social and economic impacts which stretched beyond the primary health crisis. The study focused on the current scenario of food security and nutritional adequacy of the households residing in rural municipalities of Lalitpur district, Nepal. A total sample size of 432 was estimated based on the single proportional formula n = $Z^2pq/d^2$; taking 23% proportion rate [18] with $\alpha$ level of significance at 5%. In the formula, Z = standard normal deviation and equaled 1.96 at $\alpha$ level of significance; p is the prevalence of the outcome of interest which was set at 0.5 considering the prevalence of household food insecurity, q = 1-p; design effect of 1.5 and both the margin of error (d) and non-response rate were set at 5% each.

### Sampling technique

We purposively selected Lalitpur district which consisted of three rural municipalities (*Gaupalika*) (lower administrative unit of Nepal) namely *Bagmati*, *Konjyosom*, and *Mahankal*. A *gaupalika* or rural municipality is one of the administrative divisions of Nepal and is a sub-unit within the district [23]. A rural municipality in Nepal is usually further divided into nine

administrative units called wards which are the lowest administrative units [22]. The total required samples (432) were equally divided across the three rural municipalities (144 in each rural municipality). From each municipality we further selected four wards using simple random sampling, such that we had to enroll 28 samples from each ward making a total of 144 samples in each of the rural municipalities. Within each ward, we calculated the sampling interval by dividing the total households in that particular ward by the required sample size within the ward i.e. 28. Following which we then started with a first household randomly and then consecutively selected household as per the interval for that particular ward to get our required number. In each household, the head of the household was interviewed. In case of the unavailability of the household head, the member who is 18 years and above was interviewed. For the purpose of conducting interviews, research assistants were selected based on their academic background and their experience in conducting such data collection measures including quantitative techniques. Adding to this, careful consideration was given to the situational context with all relevant safety precaution guidelines being strictly maintained and followed throughout the period of data collection.

## Data collection procedure

Four trained research assistants were selected and tasked to collect information on the socio-demographic characteristics of the participants; the effect of COVID-19 on their income and livelihood, household's access to food and dietary diversity. A pretested structured questionnaire was used for the interviews. Questionnaires were translated from English to Nepali. Nepali version of the questionnaire was pretested among 10% of the study sample (n = 44) in neighboring areas before the tools were used for data collection.

## Socio-demographic characteristics

The socio-economic variables were used to assess the effect of COVID-19 on Household Food Insecurity Access Scale (HFIAS) and HDDS which comprised of age, gender, ethnicity, level of education, occupation, monthly income, family type, job loss, income loss, and the support they have been receiving from the government and concerned stakeholders throughout the entirety of the pandemic. Basically, *Chhetri* and *Brahmin* were classed as an advantaged ethnic group whereas participants of household other than them were grouped as disadvantaged ethic community. Presence of major physical and mental disabilities among the members of household selected for this study was assessed by asking participants on any such known health adversities among the members of their household. Furthermore, presence of chronic illness in the members of household was assessed using "Yes" or "No" question. Also, job losses as well as income loss due to COVID-19 were assessed to highlight its impact on rural people. However, to account for the support provided this study only entails a broader approach in assessing the support received by the participants from various sources including government, non-governmental organizations (NGOs), political parties, social workers and doesn't reflect individual effort.

## Household Food Insecurity Access Scale

Household food insecurity access was measured using guidelines of Food and Agriculture Organization-Food And Nutrition Technical Assistance (FAO-FANTA) adopting HIFAS tool. Household food insecurity access was then illustrated using the indicator of Household Food Insecurity Access Prevalence (HFIAP) Status whereby household food insecurity access was assessed into three categories including food security, mild-to-moderate food insecurity, and severe food insecurity using HFIAS indicator guide [5, 24].

## Household Dietary Diversity Score

The HDDS were collected using a 24-hour dietary recall developed by FAO and the FANTA Project [19]. A total of 12 food groups were included in HDDS. The total dietary diversity score (DDS) ranges from 0 to 12. Food groups consumed during the previous 24 hours by the households were given a point score yielding a maximum total DDS of 12 points if his/her responses were positive to all food groups. Further, the HDDS scores were categorized as low DDS with 0–3 food groups, moderate DDS with 4–6 food groups, and high DDS with 7–12 food groups consumed by the members of that particular households during those reference periods [19, 24].

## Data analysis and management

Data compilation, checking, and coding were carried out following the data collection. Data were systematically coded and entered into Epi Data 3.1. The entered data was exported to Statistical Package for Social Sciences (SPSS) Version 20 and checked for its consistency. All analysis was finally performed using SPSS version 20. Descriptive statistics (frequency, mean and standard deviation) were presented in a frequency table. Inferential statistics such as chi-square test was applied to test the significance of the association between independent and dependent variables. For each outcome variable and independent variable, using a binary logistic regression model, a bivariate analysis was performed to assess the association between independent and outcome variables. P-values <0.05 were considered statistically significant. Based on the findings of bivariate analyses, the model for multivariable analysis was decided using all those with significant associations in the bivariate analyses. In order to account for potential confounders, independent variables that had a p-value less than 0.2 were also included in the multivariable analysis. To prevent statistical bias in the multivariable logistic regression model, we examined multi co-linearity among the independent variables using variation inflation factors (VIF). We used "10" as a cut-off value for the maximum level of VIF. Results are presented as crude odds ratio (cOR) and adjusted odds ratio (aOR) with 95% confidence intervals (CIs).

## Ethical approval

Ethical approval was obtained from the Ethical Review Board (ERB) of the Nepal Health Research Council (NHRC; Ref: 2556). Participants were detailed about the study following which both verbal and written consent from the participants was taken before conducting the survey. In case of participants with no education, verbal consent followed by thumbprint was collected as an approval for the enrollment in the study. Also, the participant's dignity was maintained by giving the right to reject or discontinue the research study at any time.

## Results

### Socio-demographic characteristics of the participants

Table 1 depicts the socio-demographic characteristic of study households in relation to food insecurity access and household dietary diversity score. Socio-demographic characteristics of the households interviewed showed the mean age (SD) of the participants to be 45 years. More than two-thirds (64%) of the total households were *Brahmin* followed by slightly more than one-fourth households (26.2%) of *Janajati*. The majority of the households were found to be Hindu (78.7%). Two-thirds of the interviewed participants were found to have attended school with one-fifth of them being able to generally read and write only. Agriculture was found to be the major source of income for almost four-fifths of the interviewed households. Only 6.3% of

**Table 1. Socio demographic characteristics and COVID related factors by household food security status and household dietary diversity status of the participants.**

| Variables | Total n (%) | HFIAS | | p-value[1] | HDDS | | p-value[1] |
|---|---|---|---|---|---|---|---|
| | | Food secure n (%) | Food insecure n (%) | | High | Low/medium | |
| | | 63.9 (95% CI: 59.2–68.3) | 36.1 (95% CI: 31.7–40.8) | | 36.8 (95% CI: 32.4–41.5) | 63.2 (95%CI: 58.5–67.6) | |
| **Socio demographic characteristics** | | | | | | | |
| **Respondent Age category** | | | | 0.001* | | | 0.939 |
| 18–40 | 146 (33.8) | 77 (27.9) | 69 (44.2) | | 53 (33.3) | 93 (34.1) | |
| 41–64 | 261 (60.4) | 185 (67.0) | 76 (48.7) | | 96 (60.4) | 165 (60.4) | |
| >64 | 25 (5.8) | 14 (5.1) | 11 (7.1) | | 10 (6.3) | 15 (5.5) | |
| **Ethnicity** | | | | <0.001* | | | 0.042* |
| Advantaged ethnic group | 313 (72.5) | 225 (81.5) | 88 (56.4) | | 123 (77.4) | 190 (69.6) | |
| disadvantaged ethnic group | 119 (27.5) | 51 (18.5) | 68 (43.6) | | 36 (22.6) | 83 (30.4) | |
| **Religion** | | | | <0.001* | | | 0.095 |
| Hindu | 340 (78.7) | 101 (64.7) | 239 (86.6) | | 132 (83.1) | 208 (76.2) | |
| Non Hindu | 92 (21.3) | 55 (35.3) | 37 (13.4) | | 27 (16.9) | 65 (23.8) | |
| **Family type** | | | | 0.853 | | | 0.943 |
| Nuclear | 183 (42.4) | 116 (42.1) | 67 (42.9) | | 67 (42.1) | 116 (42.5) | |
| Extended | 249 (57.6) | 160 (57.9) | 89 (57.1) | | 92 (57.9) | 157 (57.5) | |
| **Education** | | | | 0.002* | | | <0.001* |
| No education | 240 (55.6) | 144 (52.2) | 96 (61.5) | | 51 (32.1) | 189 (69.2) | |
| Primary | 146 (33.8) | 92 (33.3) | 54 (34.6) | | 79 (49.7) | 67 (24.5) | |
| Higher secondary or above | 46 (10.6) | 40 (14.5) | 6 (3.9) | | 29 (18.2) | 17 (6.2) | |
| **Occupation** | | | | <0.001* | | | <0.001* |
| Agriculture | 343 (79.4) | 199 (72.1) | 144 (92.3) | | 111 (69.8) | 132 (84.9) | |
| Service | 43 (9.9) | 38 (13.8) | 5 (3.2) | | 27 (16.9) | 16 (5.8) | |
| Business | 46 (10.6) | 39 (14.1) | 7 (4.5) | | 21 (13.2) | 25 (9.3) | |
| **Household head** | | | | 0.432 | | | 0.420 |
| Male | 399 (92.4) | 257 (93.1) | 142 (91.1) | | 149 (93.7) | 250 (91.6) | |
| Female | 33 (7.6) | 19 (6.8) | 14 (8.9) | | 10 (6.3) | 23 (8.4) | |
| **Source of income** | | | | <0.001* | | | 0.146 |
| Agriculture | 334 (77.3) | 199 (72.1) | 135 (86.5) | | 115 (72.3) | 219 (80.2) | |
| Business | 54 (12.5) | 47 (17.1) | 7 (4.5) | | 23 (14,5) | 31 (11.4) | |
| Service | 44 (10.2) | 30 (10.8) | 14 (9.0) | | 21 (13.2) | 23 (8.4) | |
| **Chronic illness** | | | | 0.001* | | | 0.292 |
| No | 374 (86.6) | 250 (90.6) | 124 (79.5) | | 149 (93.7) | 248 (90.8) | |
| Yes | 58 (13.4) | 26 (9.4) | 32 (20.5) | | 10 (6.3) | 25 (9.2) | |
| **COVID related factors** | | | | | | | |
| **Labor migrant abroad** | | | | 0.333 | | | 0.292 |
| No | 397 (91.9) | 251 (90.9) | 146 (93.6) | | 149 (93.7) | 248 (90.8) | |
| Yes | 35 (8.1) | 25 (9.1) | 10 (6.4) | | 10 (6.3) | 15 (9.2) | |
| **Labor migrant returned before COVID** | | | | 0.302 | | | 1.000 |
| No | 428 (99.1) | 272 (98.5) | 156 (100) | | 158 (99.4) | 270 (98.9) | |
| Yes | 4 (0.9) | 4 (1.5) | 0 | | 1 (0.6) | 3 (1.1) | |
| **Remittance during COVID** | | | | 1.000 | | | 0.337 |
| No | 422 (97.7) | 269 (97.5) | 153 (98.1) | | 157 (98.7) | 265 (97.1) | |
| Yes | 10 (2.3) | 7 (2.5) | 3 (1.9) | | 2 (1.3) | 8 (2.9) | |

(*Continued*)

**Table 1.** (Continued)

| Variables | Total n (%) | HFIAS | | p-value[1] | HDDS | | p-value[1] |
|---|---|---|---|---|---|---|---|
| | | Food secure n (%) | Food insecure n (%) | | High | Low/medium | |
| | | 63.9 (95% CI: 59.2–68.3) | 36.1 (95% CI: 31.7–40.8) | | 36.8 (95% CI: 32.4–41.5) | 63.2 (95%CI: 58.5–67.6) | |
| **Job loss due to COVID** | | | | 1.000 | | | 1.000 |
| No | 428 (99.0) | 273 (98.9) | 155 (99.4) | | 158 (99.4) | 270 (98.9) | |
| Yes | 4 (1.0) | 3 (1.1) | 1 (0.6) | | 1 (0.6) | 3 (1.1) | |
| **Income loss due to COVID** | | | | 1.000 | | | 1.000 |
| No | 430 (99.5) | 275 (99.6) | 155 (99.4) | | 158 (99.4) | 272 (99.6) | |
| Yes | 2 (0.5) | 1 (0.4) | 1 (0.6) | | 1 (0.6) | 1 (0.4) | |
| **Government support** | | | | 0.034 | | | 0.311 |
| No | 327 (75.7) | 218 (78.9) | 109 (69.9) | | 116 (72.9) | 211 (77.3) | |
| Yes | 105 (24.3) | 58 (21.1) | 47 (30.1) | | 43 (27.1) | 62 (22.7) | |
| **COVID support** | | | | 0.638 | | | 0.858 |
| No | 293 (67.8) | 185 (67.1) | 108 (69.2) | | 107 (67.3) | 186 (68.1) | |
| Yes | 139 (32.7) | 91 (32.9) | 48 (30.8) | | 52 (32.7) | 87 (31.9) | |

[1] Chi square test or Fischer exact test

*statistically significant at p<0.05; HFIAS: Household Food Insecurity Access Scale; HDDS: Household dietary diversity score

the participating households reported having major physical and mental disabilities whereas 13.4% of them reported the presence of chronic diseases in their household (Table 1).

## Status of household's food security access

Results of the assessment of household's food security status revealed that more than three-fifth [64% (95% CI: 59.2–68.3)]of the households reported being food secure followed by more than one-third [36.1% (95% CI: 31.7–40.8)] being food insecure whereby, nearly one-fourth being mildly food insecure, exactly one-tenth being moderately food insecure, and only 3% being severely food insecure, respectively (**Fig 1**).

# Status of household's food security access

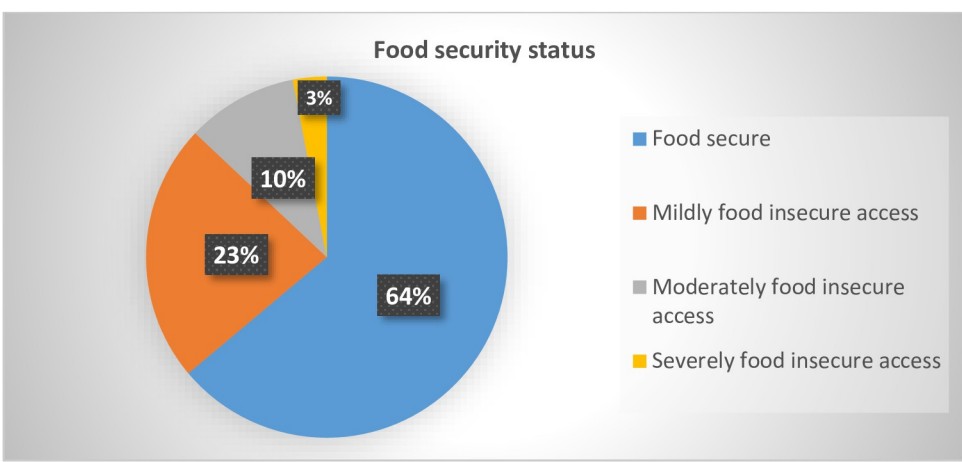

**Fig 1. Household's food security status.**

Our findings further revealed that among those who were food insecure, 44.2% of the household's respondents were of age category18–40 years and 49% of them were of age 41–64 years respectively. Similarly in terms of ethnicity of food insecure households, more than half (56%) of the advantaged ethnic groups were found food insecure followed by nearly 44% food insecurity amongst household from disadvantaged community. Likewise, 61.5% of the households with no parental education were food insecure followed-up by 34.6% with parents having primary education. Our results also revealed that the source of income was also one of the detrimental factors for food security. Household with agriculture as their primary source of income were found food secure compared to their counterpart household with business and service as their major source of income.

The HDDS also followed similar pattern. It showed 36.8% (95% CI: 32.4–41.5) of the households to have high dietary diversity score, whereas 63.2% (95%CI: 58.5–67.6) had low to medium dietary diversity score. Among the participants consuming low/medium dietary diversity, 69.6% were from advantaged ethnic group and remaining 30.4% were from disadvantaged ethnic group. Parental education also found to be significantly associated with HDDS. While taking low/Medium dietary diversity score into account, those with no education accounted for 69.2% of the households whereas, those with primary education and higher secondary/higher education accounted for 24.5%, and 6.2% of the households respectively.

## Association of socio-demographic and COVID related variables with household dietary diversity and household food insecurity score

Household having participants aged 41–64 years had a 65% lower odds of HFIAS (aOR = 0.35, 95% CI: 0.21–0.59) and 52% lower odds of having low/medium dietary diversity (aOR = 0.48, 95% CI: 0.28–0.83) compared to those household having participants aged 18–40 years. By ethnicity, the disadvantaged ethnic group had nearly three times higher odds of HFIAS (aOR = 2.73, 95% CI: 1.23–6.07). compared to advantaged ethnic groups. Participants who reported having no education were associated with increased scores on both the HFIAS (aOR = 3.70, 95% CI: 1.16–11.71) and HDDS (aOR = 10.05, 95% CI: 4.05–24.91) compared to the head of a household having higher secondary and above. Also, households having patients suffering from chronic illness (aOR = 3.12, 95% CI: 1.53–6.35) were associated with higher odds of HFIAS. Regarding the COVID-related characteristics, in an unadjusted analysis, households reported receiving government support during the COVID-19 pandemic had a significantly positive association with HFIAS score (cOR = 1.62, 95% CI: 1.03–2.53) compared to those who did not receive government support during COVID-19 pandemic (Table 2).

## Discussion

With the COVID-19 pandemic emergence, it is clear that not only the multiple health aspects of people around the world have been affected but also escalated the level of their food insecurity [7, 25–28]. Although, healthy and balanced food intake is crucial concerns regarding the surging food insecurity are arising. There exists a lack of evidence regarding household food insecurity and the nutritional adequacy of the people living in rural communities during the emergency context of the pandemic. To bridge the existing gap, this study highlights the prevalence of household food insecurity, dietary diversity, and associated factors during the COVID-19 lockdown among rural households of Lalitpur district, Nepal.

Our study demonstrated the high prevalence of food insecurity with 36% of the study participants experiencing some degree of food insecurity which is 13% higher compared to the national average reported by previous study conducted during COVID among households of all seven provinces of the country [18]. This finding is consistent with a study from the

**Table 2. Factors associated with household food security and insecurity status and household dietary diversity status of the participants.**

| Variables | HFIAS | | HDDS | |
|---|---|---|---|---|
| | cOR (95% CI) | aOR (95% CI)[2] | cOR (95% CI) | aOR (95% CI)[2] |
| **Socio demographic characteristics** | | | | |
| **Respondent Age category** | | | | |
| 18–40 | Ref | Ref | Ref | Ref |
| 41–64 | 0.45 (0.30–0.69)*** | 0.35 (0.21–0.59)*** | 0.97 (0.64–1.49) | 0.48 (0.28–0.83)** |
| >64 | 0.87 (0.37–2.05) | 0.22 (0.07–0.66)** | 0.85 (0.35–2.03) | 0.33 (0.11–1.03) |
| **Ethnicity** | | | | |
| Advantaged ethnic group | Ref | Ref | Ref | Ref |
| disadvantaged ethnic group | 3.40 (2.19–5.28)*** | 2.73 (1.23–6.07)* | 1.49 (1.17–2.34)* | 1.27 (0.56–2.84) |
| **Religion** | | | | |
| Hindu | Ref | Ref | Ref | Ref |
| Non Hindu | 3.51 (2.18–5.66)*** | 2.00 (0.84–4.73) | 1.52 (0.92–2.51) | 1.02 (0.42–2.45) |
| **Family type** | | | | |
| Nuclear | Ref | Ref | Ref | Ref |
| Extended | 0.96 (0.64–1.43) | 0.95 (0.59–1.55) | 0.98 (0.66–1.46) | 0.91 (0.57–1.45 |
| **Education** | | | | |
| No education | 4.44 (1.81–10.88)** | 3.70 (1.16–11.71)* | 6.32 (3.22–12.40)*** | 10.05 (4.05–24.91)*** |
| Primary | 3.91 (1.55–9.83)** | 3.67 (1.23–9.89)* | 1.44 (0.73–2.85) | 1.68 (0.76–3.73) |
| Higher secondary or above | ref | Ref | Ref | Ref |
| **Occupation** | | | | |
| Agriculture | Ref | Ref | Ref | Ref |
| Service | 0.18 (0.06–0.47)*** | 0.22 (0.06–0.77)* | 0.28 (0.14–0.54)*** | 0.70 (0.26–1.84) |
| Business | 0.24 (0.10–0.57)*** | 0.47 (0.13–1.87) | 0.56 (0.30–1.06) | 0.78 (0.25–2.39) |
| **Household head** | | | | |
| Male | Ref | Ref | Ref | Ref |
| Female | 1.33 (0.64–2.74) | 1.03 (0.42–2.51) | 1.37 (0.63–2.95) | 0.93 (0.37–2.32) |
| **Primary source of income** | | | | |
| Agriculture | Ref | Ref | Ref | Ref |
| Business | 0.21 (0.09–0.50)*** | 0.31 (0.08–1.10) | 0.70 (0.39–1.27) | 1.00 (0.35–2.86) |
| Service | 0.68 (0.35–1.34) | 1.37 (0.52–3.52) | 0.57 (0.29–0.97)* | 1.32 (0.54–3.22) |
| **Chronic illness** | | | | |
| No | Ref | Ref | Ref | Ref |
| Yes | 2.48 (1.41–4.34)** | 3.12 (1.53–6.35)** | 0.84 (0.38–1.81) | 0.64 (0.32–1.28) |
| **COVID related factors** | | | | |
| **Labor migrant abroad** | | | | |
| No | Ref | Ref | Ref | Ref |
| Yes | 0.68 (0.32–1.47) | 0.47 (0.17–1.27) | 1.50 (0.70–3.21) | 1.36 (0.53–3.47) |
| **Labor migrant returned before COVID** | | | | |
| No | Ref | | Ref | Ref |
| Yes | 0.68 (0.32–1.47) | | 2.36 (0.49–11.30) | 1.23 (0.09–16.09) |
| **Remittance during COVID** | | | | |
| No | Ref | Ref | Ref | Ref |
| Yes | 0.75 (0.19–2.95) | 1.44 (0.24–8.55) | 1.75 (0.18–17.02) | 3.45 (0.43–27.18) |
| **Job loss due to COVID** | | | | |
| No | Ref | Ref | Ref | Ref |
| Yes | 0.58 (0.06–5.69) | 1.04 (0.06–16.77) | 1.75 (0.18–17.02) | 2.34 (0.16–33.29) |
| **Income loss due to COVID** | | | | |

*(Continued)*

**Table 2.** (Continued)

| Variables | HFIAS | | HDDS | |
|---|---|---|---|---|
| | cOR (95% CI) | aOR (95% CI)[2] | cOR (95% CI) | aOR (95% CI)[2] |
| No | Ref | Ref | Ref | Ref |
| Yes | 1.77 (0.11–28.56) | 0.61 (0.02–18.75) | 0.47 (0.04–4.59) | 0.15 (0.03–6.42) |
| **Government support** | | | | |
| No | Ref | Ref | Ref | Ref |
| Yes | 1.62 (1.03–2.53)* | 1.47 (0.84–2.56) | 1.30 (0.66–2.55) | 0.72 (0.40–1.31) |
| **COVID support** | | | | |
| No | Ref | Ref | Ref | Ref |
| Yes | 0.90 (0.59–1.37) | 0.72 (0.43–1.19) | 0.83 (0.47–1.48) | 1.10 (0.68–1.78) |
| **HDDS** | | | | |
| High | Ref | Ref | | |
| Low and medium | 1.15 (0.47–0.76) | 0.81 (0.48–1.35) | | |
| **HFIAS** | | | | |
| Food secure | | | Ref | Ref |
| Mildly food secure | | | 0.98 (0.31–3.09) | 0.92 (0.24–3.53) |
| Moderately food secure | | | 1.12 (0.71–1.77) | 0.78 (0.44–1.37) |
| Severely food secure | | | 1.41 (0.64–3.09) | 0.80 (0.32–1.96) |

*p<0.05

**p<0.01

***p<0.001; cOR = crude odds ratios for unadjusted model; aOR = adjusted odds ratios for adjusted model; CI: confidence interval; Ref: reference category; [2]Single model was run for adjusting the variables p<0.2 in the unadjusted model

Pakistan whereby 36.3% participants were facing some degree of food insecurity [29]. COVID validates its impact on food security with reported 32.4% households assessed to have significant food insecurity status [30]. Also, our study found that households having patients suffering from chronic illness were associated with higher odds of HFIAS. This is supported by the evidence generated from multiple studies around the world especially linking with the presence of diabetes and markers of other chronic diseases [31–34]. This might be because of the fact that food insecure household have increased dependency on inexpensive, highly palatable foods that are energy dense leading to the development of chronic conditions [35].

In addition, the effects of COVID-19 on household food security status were observed in a research conducted in Bangladesh which illustrated the rise of food insecurity from 45% to 61% [36]. Aligning with the evidence generated from the researches worldwide, our result pinpoints the further aggravated scenario of food insecurity among households which can be attributed to COVID-19 and immediate cautionary practices adopted for prevention [10, 36]. Throughout COVID-19, food insecurity among low-income and disadvantaged families in Nepal significantly affected their health and well-being [8], whereby 7.4% households reported on adopting negative livelihood coping strategies to address food shortage [18]. These observed adverse scenario might have been influenced by the nationwide lockdown and subsequent restriction of socioeconomic activities all over the country [36].

Also, the results obtained from our study are consistent with the results obtained from various researches conducted targeting similar objectives [3, 6, 27]. Results of this study showed that the participants who reported having no education were associated with increased scores on both the HFIAS and HDDS compared to the head of a household with an educational level of having higher secondary and above. Similarly, study conducted in Iran highlighted that the

participants with the knowledge about nutrition from health professionals and other sources had greater HDDS and HFIAS compared to the non-educated group which our study completely disagrees with [37]. Even the previously mentioned study conducted by the same researchers within similar settings concluded inconsistent results whereby education was found to have influential role in having better food insecurity access scores [5].

Our result showed age as the predictor factor as participants aged 41–64 years were less likely to have household food insecurity and consumed diverse diet respectively. However, contrasting results were concluded by researches conducted during the context of COVID [38, 39]. This clearly contradict the statement made by the research conducted by Abdullah et al. whereby household with older participants as the head of the family were found to have high food insecurity [40]. Fascinatingly enough, this study revealed that the respondent's household who reported receiving support (of any form; either money or ration) during the COVID-19 pandemic had a significantly positive association with HFIAS score compared to those who did not receive any form of support from the government and concerned stakeholders. Despite the high household food insecurity, livelihood as well as income loss reported by WFP [17], the support provided whether cash or food material, was a one-time thing whereas on the other hand pandemic scenario and the lockdown lasted for an extensive period of nearly two and a half years [41]. These factors must have played crucial role in deliberately pushing vulnerable people not involved in any form of agricultural activities to the edge.

Despite the concerning high levels of food insecurity, household dietary diversity score (HDDS>4) was low/medium for 63% of households in the study population and high for 36.8% of households. No such association was observed which implied the COVID-19 related impact namely job loss, and income loss on HDDS which wasn't the case in various similar studies [42, 43]. This might be contributed by the involvement in agricultural activities of the majority of participants residing in that particular region regardless of other activities contributing to their income generation.

Our study had some strength. At first, this survey evaluated the effect of COVID-19 pandemic on households' food insecurity and dietary diversity in people residing in rural areas of Lalitpur district Nepal using a household survey through face-to-face interviews. This in turn gave us a clear cut view of a large number of socioeconomic characteristics associated with food insecurity access and dietary diversity during the emergencies. Incorporating dietary diversity of rural residents of Nepal as one of the significant objective along with the large sample size included is also the strength of this study. Also, our study is expected to serve as a piece of evidence to improve nutritional status of people residing in rural community by intervening with the food security and dietary diversity during the emergencies of such scale. With that being mentioned, this study has not beyond some limitations. Firstly, as this study was conducted in three rural municipalities of one district of Nepal so may not be generalized to every rural settings of this country. Also, the study design implicated precludes any possible relationship between the predictors and the outcome (food insecurity, dietary diversity). Moreover, the dietary diversity score is calculated solely being based on 24-hour food consumption and was entirely based on subjective perceptions and hence could be subjected to recall bias.

## Conclusion

This study showed substantial prevalence of household food insecurity among people residing in rural areas in the vicinity of Kathmandu valley, Nepal; mainly among the disadvantaged ethnic groups and people having lower educational level during the COVID-19 pandemic highlighting serious public health concerns. Despite the raised level of food insecurity, dietary diversity was found satisfactory as almost every household was found to have at least low to

medium dietary diversity during per day course of their meal. This reflects a need of attention on food insecurity from the concerned stakeholders targeting the disadvantaged ethnic group and people having lower educational level. Timely and tailored response from the government focusing on these particular groups of people might help in improving HFIAS and HDDS among the people living in similar settings during emergency situation.

## Supporting information

**S1 Fig. Conceptual framework on household food security access and dietary diversity amidst COVID-19 pandemic.**
(TIF)

## Acknowledgments

First and foremost authors would like to thank Public Health Promotion and Development Organization for their technical assistance throughout the study period required for the successful completion of this study. Also, the authors would like to express their gratitude to the participants of this study who provided the information related to this study and helped to make our work successful. Without this, the study would not have been possible.

## Author Contributions

**Conceptualization:** Dirghayu K. C., Namuna Shrestha.

**Data curation:** Dirghayu K. C.

**Formal analysis:** Dirghayu K. C., Namuna Shrestha, Dev Ram Sunuwar.

**Methodology:** Dirghayu K. C., Namuna Shrestha, Rachana Shrestha, Dev Ram Sunuwar, Anil Poudyal.

**Project administration:** Dirghayu K. C.

**Supervision:** Dirghayu K. C., Namuna Shrestha, Rachana Shrestha, Dev Ram Sunuwar, Anil Poudyal.

**Validation:** Dirghayu K. C., Namuna Shrestha, Rachana Shrestha, Dev Ram Sunuwar, Anil Poudyal.

**Writing – original draft:** Dirghayu K. C., Namuna Shrestha, Anil Poudyal.

**Writing – review & editing:** Dirghayu K. C., Namuna Shrestha, Rachana Shrestha, Dev Ram Sunuwar, Anil Poudyal.

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
