## [Decision Letter · Decision Letter 0]

24 Nov 2022

PONE-D-22-20978Title-Household Food Security access and Dietary Diversity amidst COVID-19 Pandemic in rural Nepal; an evidence from rapid assessmentPLOS ONE

Dear Dr. K.C.,

Thank you for submitting your manuscript to PLOS ONE. After careful consideration, we feel that it has merit but does not fully meet PLOS ONE’s publication criteria as it currently stands. Therefore, we invite you to submit a revised version of the manuscript that addresses the points raised during the review process.

We look forward to receiving your revised manuscript.

Kind regards,

Dr. Md Nazirul Islam Sarker

Academic Editor

PLOS ONE

Journal Requirements:

No. The funders had no role in study design, data collection and analysis, decision to publish, or preparation of the manuscript. 

Reviewers' comments:

Reviewer's Responses to Questions

**Comments to the Author**

1. Is the manuscript technically sound, and do the data support the conclusions?

Reviewer #1: Partly

Reviewer #2: Yes

2. Has the statistical analysis been performed appropriately and rigorously? 

Reviewer #1: No

Reviewer #2: Yes

3. Have the authors made all data underlying the findings in their manuscript fully available?

Reviewer #1: Yes

Reviewer #2: Yes

4. Is the manuscript presented in an intelligible fashion and written in standard English?

Reviewer #1: No

Reviewer #2: Yes

5. Review Comments to the Author

Reviewer #1: This paper provides a cross-sectional analysis on food security prevalence and associated household factors in purposively selected 'rural' subdistricts in Lalitpur district, Nepal during one of the COVID-19 lockdown periods in 2021. The broader objective of the study - to provide a snapshot of food security, household dietary diversity and associated factors during a pandemic, while important, was not done full justice in the preparation of the manuscript and its results. The presentation of the study design, results and discussion while mostly defensible have not been adequately presented as such. My suggestion is that a full and detailed revision be completed prior to consideration for publication. I hope the comments provided below may provide some helpful input on how to possibly clarify the detailing of the study and results for a reader.

Overall, there are issues of grammatical errors and language clarity and the paper would benefit from an editorial review.

Abstract: The study design is described as "analytical cross-sectional", what do the authors mean by analytical? This is not common utilized terminology to describe study design. Next, what are determining factors? Please clarify. In the methods section, please make mention of what models were used. For the results, reported AORs should come at the end of the sentence for readability and accuracy: "Multiple regression showed the disadvantaged ethnic group (AOR=2.73, 95% CI: (1.23-6.07) who did not attend the formal education (AOR= 3.70, 95% CI: 1.16-11.71) had significant higher odds of household food insecurity. Likewise, participants with no formal education (AOR=, 95% CI: 10.05 (4.05-24.91) were more likely to have household food insecurity." What are ref groups? What is the HFS score? The conclusion speaks of exacerbation but no basis of this conclusion.Als says dietary diversity to acceptable but no average for study was provided.

Data access: Where is the data for this study made available?

Line 69-70 - what is the % increase?

Line 73 - what is relevant food insecurity? Unclear.

Line 73: Mentions that 4.6 million people were food insecure - when? If the argument being made is that food insecurity is likely worse, would be helpful to have more of a timeline setup. Recommend changing suffer to experience based on how food security is assessed.

Line 78-79 - If you choose to make mention of province 2, would include language to indicate what makes province 2 important i.e. that you're speaking of within country variation where some provinces are worse off than others. Otherwise, for a wider audience, province 2 has little relevance. Overall, I recommend giving a little more contextual explanation as it relates to social groups (lots of mention of disadvantaged groups, etc), sampling units.

Corona virus disease and social cultural are not commonly used terms to describe the virus and socio-cultural factors, respectively.

Unclear what the sentence that runs on into lines 83-84 means.

Line 98-99 - there are other factors that affect food insecurity in rural areas and if we are talking about access and availability it is not just related to a dependence of rain but instead general low ag productivity, a lack of ag innovations, inadequate market access and so forth.

Methods: Why did the authors decide on Lalitpur as a study site given the interest in rural areas that might be best studied elsewhere in the country? How many urban vs rural municipalities in Lalitpur are there? What was the rationale for Bagmati province? Please provide clearly the primary sampling unit and detail the systematic random sampling method more elaborately. What are quantitative techniques? Please provide explanation of what gaupalikas are, what was the inclusion criteria was for the study and who specifically in the household was interviewed (if household head, provide definition of household head). I recommended this last part as throughout the methods, results and discussion, the authors weave in and out of household versus individual level language.

For the analysis of factors related to food insecurity and household dietary diversity, what informed the choice of the factors used in the analysis? What theoretical or conceptual underpinning did the authors have to include the factors they did in the analysis?

Line 145: How was family members use to assess impact of COVID 19 The use of the word 'impact' may ve too strong of language given what the authors are referring to. There are discrepancies between the what data is reported to be collected during the interviews and what comes into the reporting further down the line in the paper (household size, etc). In line 159-165 explain household consumption vs individual because as it reads now, it is not clear.

The authors report using semi-structured interviews - what was the unstructured portion of the interview? What were the type of questions were asked?

Line 188: add specifics, participants were heads of hh

Tables not formatted in a reader friendly format on page 17. Most of the tables to would benefit from some major reformatting.

Lines 193-194: What is the sig of the caste and religious groupings - would encourage more detail to be provided on this to help with interpretation. In the methods section, the assessment of medical history and disabilities not included in methods - line 193- 194

What is driving the age categorization cut offs?

Results: The results need to be more clearly explained with more frequent mention of reference groups all the results being discussed are compared to.

Discussion:

There is very little discussion on any other food security data from Nepal during the COVID-19 pandemic nor contextual information about what characteristics of your study areas might be feeding into the results you are seeing. This is not a representative sample of Lalitpur nor a typical rural area of Nepal - if the authors disagree, would urge them to present their argument of why clearly when they describe the strengths of the study .

Line 274: Would be very careful about using causal language in a cross-sectional paper of non-representative population.

Reviewer #2: I appreciate all authors for this study. Although the study's research is quite unique, the authors should provide more related existing literature. Furthermore, the author didn`t clear properly why and this study is very relevant and significant and what is the policy implications? Please address this issue properly. The author(s) can also use a table to present a review of the literature. Determine the research gap in previous literature and how this work differs from previous efforts. Mention the gap in research. However, the methodology is clear and good.

The author can focus to create link how the results on this study fit in with the results from previous studies. The conclusion section is so weak, can provide summary of findings in this part. Author should discuss about research strength and limitations if have in this section. Lastly, I strongly suggest to give a paragraph about policy recommendations based on findings.

6. PLOS authors have the option to publish the peer review history of their article (what does this mean?). If published, this will include your full peer review and any attached files.

Reviewer #1: No

Reviewer #2: No

---

## [Author Response · Author response to Decision Letter 0]

16 Jan 2023

15 January 2023

The Academic Editor,

PLOS ONE 

Subject: Resubmission of the research article entitled “Household Food Security access and Dietary Diversity amidst COVID-19 Pandemic in rural Nepal; an evidence from rapid assessment" (Manuscript ID: PONE-D-22-20978)

Dear Editor and Reviewers,

It is our great pleasure to submit our response to the reviewers comments on our manuscript entitled "Household Food Security access and Dietary Diversity amidst COVID-19 Pandemic in rural Nepal; an evidence from rapid assessment” to PLOS ONE. We have tried our best to address comments from both the Reviewers and editor. Thank you for your concern regarding these issues. Here, we have highlighted our point-by-point comments and the accompanying adjustments to the manuscript. We would be looking forward to your response and any further correction to be made to enrich as well as improve the quality of this paper.

We look forward to your kind consideration,

Corresponding author

Dirghayu KC

Public Health Promotion and Development Organization

Response to #Reviewer 1

Reviewer #1: This paper provides a cross-sectional analysis on food security prevalence and associated household factors in purposively selected 'rural' sub districts in Lalitpur district, Nepal during one of the COVID-19 lockdown periods in 2021. The broader objective of the study - to provide a snapshot of food security, household dietary diversity and associated factors during a pandemic, while important, was not done full justice in the preparation of the manuscript and its results. The presentation of the study design, results and discussion while mostly defensible have not been adequately presented as such. My suggestion is that a full and detailed revision be completed prior to consideration for publication. I hope the comments provided below may provide some helpful input on how to possibly clarify the detailing of the study and results for a reader.

Overall, there are issues of grammatical errors and language clarity and the paper would benefit from an editorial review.

Response: Thank you for your constructive feedbacks. Authors really appreciate the effort of reviewer in reviewing this study. The whole manuscript has been checked and corrected for grammatical errors as well as language clarity. 

Abstract: The study design is described as "analytical cross-sectional", what do the authors mean by analytical? This is not common utilized terminology to describe study design. Next, what are determining factors? Please clarify. In the methods section, please make mention of what models were used. For the results, reported AORs should come at the end of the sentence for readability and accuracy: "Multiple regression showed the disadvantaged ethnic group (AOR=2.73, 95% CI: (1.23-6.07) who did not attend the formal education (AOR= 3.70, 95% CI: 1.16-11.71) had significant higher odds of household food insecurity. Likewise, participants with no formal education (AOR=, 95% CI: 10.05 (4.05-24.91) were more likely to have household food insecurity." What are ref groups? What is the HFS score? The conclusion speaks of exacerbation but no basis of this conclusion. Also says dietary diversity to acceptable but no average for study was provided.

Response: Thank you for your suggestions. The word ‘analytical’ has been removed from the term ‘analytical cross-sectional’ and defined as cross-sectional descriptive study in the revised manuscript. By determining factors, we mean to say the factors that are found to influence household food security status and dietary diversity score. For example, in our study we found that ethnicity, education and age were the factors found to influence household food security status. We performed bivariate and multivariate analyses. Mention of the models used during our statistical analysis has been made in the method section both in the abstract and the main body of the manuscript. We have explained with further clarity about how variables were chosen for the model. The suggested changes for reporting AOR have been made. The reference groups have been denoted as “Ref” in Table 3 which was used to compute the odds ratio in the regression model. To give example, for disadvantaged ethnic group, advantaged ethnic group are the reference group. We have calculated the proportion of people who are food secure and unsecured as64% and 36% respectively. The conclusion in the abstract has been revised according to the findings of the study and we have modified the limitation section such that the readers are aware of what the study limitations were. We have also adjusted the manuscript further as suggested by the reviewer. As recommended, the whole result section has been rearranged and represented.

Mention of the models used during our statistical analysis has been made in the method section both in the abstract and the main body of the manuscript. We have explained with further clarity about how variables were chosen for the model. We have modified the conclusion to match the findings of the study, and also have modified the limitation section such that the readers are aware of what the study limitations were. Reference group has been mentioned in results wherever necessary for clarity. 

Data access: Where is the data for this study made available?

Response: After publication of this study, data will be made available by authors upon request.

Line 69-70 - what is the % increase?

Response: Authors would like to thank reviewer for pointing out this issue. The whole sentence has been rephrased as well as the information provided in the revised manuscript. Reference to the statement has also been adjusted subsequently. 

Line 73 - what is relevant food insecurity? Unclear.

Response: Thank you for your comment. The word ‘relevant’ was extra in the sentence and has been removed from the sentence in the revised version of the manuscript to make it clearer.

Line 73: Mentions that 4.6 million people were food insecure - when? If the argument made is that food insecurity is likely worse, would be helpful to have more of a timeline setup. Recommend changing suffers to experience based on how food security is assessed.

Response: Thank you for your comment. Data representing the scenario of food insecurity in Nepal have been revised and presented along with the year. Changes suggested by the reviewers have been addressed in the revised manuscript. 

Line 78-79 - If you choose to make mention of province 2, would include language to indicate what makes province 2 important i.e. that you're speaking of within country variation where some provinces are worse off than others. Otherwise, for a wider audience, province 2 has little relevance. Overall, I recommend giving a little more contextual explanation as it relates to social groups (lots of mention of disadvantaged groups, etc), sampling units.

Response: Thank you for your comments and constructive suggestions. However, authors felt that this sentence bears less resemblance in this paragraph. Hence this sentence has been removed completely.

Corona virus disease and social cultural are not commonly used terms to describe the virus and socio-cultural factors, respectively.

Response: Thank you for pointing this out. The word mentioned by the reviewer has been changed into much commonly used term. The word ‘Corona virus disease’ has been changed into just ‘Corona virus’ and the word the word ‘social cultural’ has been changed into ‘socio-cultural’ in the revised manuscript.

Unclear what the sentence that runs on into lines 83-84 means.

Response: This sentence has been paraphrased into simple and clear ones. Thank you.

Line 98-99 - there are other factors that affect food insecurity in rural areas and if we are talking about access and availability it is not just related to a dependence of rain but instead general low ag productivity, a lack of ag innovations, inadequate market access and so forth.

Response: Thank you for your comment. A sentence has added there to highlight the context pointed out by the reviewer and presented as per the reviewer’s suggestions.

Methods: Why did the authors decide on Lalitpur as a study site given the interest in rural areas that might be best studied elsewhere in the country? How many urban vs rural municipalities in Lalitpur are there? What was the rationale for Bagmati province? Please provide clearly the primary sampling unit and detail the systematic random sampling method more elaborately. What are quantitative techniques? Please provide explanation of what gaupalikas are, what was the inclusion criteria were for the study and who specifically in the household was interviewed (if household head provide definition of household head). I recommended this last part as throughout the methods, results and discussion, the authors weave in and out of household versus individual level language.

Response: There are a total of six municipalities in Lalitpur district. Of these, there is one metropolitan city, two of them are urban municipalities and remaining three are rural municipalities. Rural municipalities of Lalitpur are; Konjyoson rural municipality, Bagmati rural municipality and Mahankal rural municipality. Being near to the capital city of Nepal, rural municipalities of Lalitpur are often neglected and have the similar socio-economic conditions as other rural municipalities of the country. Because of this scenario, authors of this study decided to select these study sites. In addition, there was limitation on budget, the authors carried out this study with a small grant that was provided by the Nepal Health Research Council. Hence, choosing the rural areas in the vicinity of Kathmandu valley provided nearly the similar picture of distant rural areas, served the purpose of the study. We have removed “Bagmati Province” as the study is more relevant to rural settings within the district and similar context. We have revisited the sampling technique part of the methodology section to elaborate more on this part. By quantitative techniques we mean to say those having experience in quantitative data collection methods such as household survey using questionnaire through face to face interview. But since we felt this much elaboration about research assistant may not be required, we have deleted that part. A gaupalika or rural municipality is one of the administrative division of Nepal and is a sub-unit within the district. A rural municipality within the district is considered as the primary sampling unit. Samples were equally divided into three rural municipalities (144 in each). Within each of the three rural municipality, 4 wards (lowest administrative unit) were selected as secondary sampling unit followed by 28 households in each of those wards. Detail description has been added to the manuscript. Within the household, particularly the head of the household and in case of unavailability of the household head, the member of the family who is 18 years and above is available comprised our study population. Head of household is the member of household who is managing household activities and takes the decisions as well as responsibility in all household related matters. 

Author would like to thank the reviewer in pinpointing such details. With regard to the comment of weaving in and out of the household versus individual level, the whole method section is being checked for the language and information flow. As suggested by the reviewer, the changes have been made in the revised manuscript.

For the analysis of factors related to food insecurity and household dietary diversity, what informed the choice of the factors used in the analysis? What theoretical or conceptual underpinning did the authors have to include the factors they did in the analysis?

Response: Thank you for your concern. Based on our literature review on household food security and dietary diversity during COVID-19 conducted prior to the study, we made choices of those factors used in the analysis. Based on the same literature review, authors of this study devised an outline including factor influencing household food insecurity and dietary diversity. Details on conceptual framework developed by the authors during proposal development of this particular study has been added in the revised manuscript and submitted as a supplementary file 1. The authors, however, did not use any particular theoretical framework to base the current analysis.

Line 145: How was family members use to assess impact of COVID 19 The use of the word 'impact' may ve too strong of language given what the authors are referring to. There are discrepancies between the what data is reported to be collected during the interviews and what comes into the reporting further down the line in the paper (household size, etc).In line 159-165 explain household consumption vs individual because as it reads now, it is not clear.

Response: Thank you for your comments. As it is already known that family type effect household food security and Dietary diversity, this has been considered one of the variables of interest to see how the size of family effects HFS and DDS of household’s being survey during COVID-19.The words ‘family members’ has been replaced with family type. The word ‘impact’ has been replaced with ‘effect’. The whole method and results section of this manuscript is checked and changes have been made to omit the existing discrepancies. The lines from 159-165 were checked for its consistent flow of information. 

The authors report using semi-structured interviews - what was the unstructured portion of the interview? What were the type of questions were asked?

Response: Authors would like to thank reviewer for comments and pointing out this mistake in our manuscript. Our interview questionnaire was completely structured. Changes have been made wherever there was mention of ‘semi-structured interview’ with ‘structured interview’ within the manuscript. 

Line 188: add specifics, participants were heads of hh

Response: Thank you for your comments. Among the participants, there were 324 heads of households. This information has been mentioned in the revised manuscript as well. 

Tables not formatted in a reader friendly format on page 17. Most of the tables to would benefit from some major reformatting.

Response: Thank you for your comment. Reformatting of table has been done by the authors to make it more reader friendly taking consideration of journal requirement of PLOS ONE.

Lines 193-194: What is the sig of the caste and religious groupings - would encourage more detail to be provided on this to help with interpretation. In the methods section, the assessment of medical history and disabilities not included in methods - line 193- 194

Response: Thank you for your comments. Caste and religious grouping performed within our study was meant to reflect the exact scenario of food security and nutritional adequacy among multiple ethnic groups residing in such rural settings. As it is a well established fact that there exists a barrier among certain caste and religious groups in accessing public services and resources in the past. In reference to these, authors of this study decided on caste and religious groupings and see whether these existed or not during the emergencies. Details on assessment of medical history and disability have been added in the socio-economic characteristics of the participant in the revised manuscript.

What is driving the age categorization cut offs?

Response: Thank you for your comments. Since our study participants primarily targeted the household head, we assumed that they may be over 40 and have defined the cut offs accordingly, and wanted to have one category for above 40 years but leaving the elderly(65 and above) in separate category, while younger population that is up to 40 years age was kept in one category. Age categorization cut-off was basically driven by the person’s role in the household and his/her involvement as well as knowledge in overall household activities of that particular household.

Results: The results need to be more clearly explained with more frequent mention of reference groups all the results being discussed are compared to.

Response: Thank you for your comments. All the results obtained in this study have been presented with the reference groups. 

Discussion:

There is very little discussion on any other food security data from Nepal during the COVID-19 pandemic nor contextual information about what characteristics of your study areas might be feeding into the results you are seeing. This is not a representative sample of Lalitpur nor a typical rural area of Nepal - if the authors disagree, would urge them to present their argument of why clearly when they describe the strengths of the study.

Response: Thank you for your comments. Authors really do acknowledge the concern of this particular reviewer. However, there were only limited article available in the field of food security and dietary diversity in Nepal during COVID-19 that we could get hold of. Further to this, there might be paucity of evidence targeting food security and dietary diversity among households of rural skirts during the pandemic. Because of these reasons, authors were bound to present limited contextual information on food security data during COVID-19. Few changes have been made and additional information on the effect of COVID-19 on livelihood of Nepalese population has been included in the discussion section of both revised manuscript and manuscript with track change.

Authors of this study agree to this particular reviewer on the fact that this study isn’t a representative sample of Lalitpur, however it may not be true that this study does not represent the typical rural area of Nepal. The rural areas of Lalitpur are truly rural and are similar to majority of the rural areas of Nepal though they seem to be near the capital city. Municipalities here in Nepal are divided into urban and rural municipalities based on multiple developmental features available there, and population that resides there within. Although, this study couldn’t represent every rural area of Nepal, it definitely provides us with the overview on the situational context of most of the rural parts of this country. All these details have been added in the revised manuscript. 

Line 274: Would be very careful about using causal language in a cross-sectional paper of non-representative population.

Response: Thank you for your valuable suggestion. We have checked the write up and replaced with a better alternative to the words reflecting causal language. 

Reviewer #2: I appreciate all authors for this study. Although the study's research is quite unique, the authors should provide more related existing literature. Furthermore, the author didn`t clear properly why and this study is very relevant and significant and what is the policy implications? Please address this issue properly. The author(s) can also use a table to present a review of the literature. Determine the research gap in previous literature and how this work differs from previous efforts. Mention the gap in research. However, the methodology is clear and good.

The author can focus to create link how the results on this study fit in with the results from previous studies. The conclusion section is so weak, can provide summary of findings in this part. Author should discuss about research strength and limitations if have in this section. Lastly, I strongly suggest to give a paragraph about policy recommendations based on findings.

Response: Authors of this study would like to thank the reviewer for comments and constructive feedbacks. We have addressed and incorporated all the suggestions made by the reviewer. We have specifically made changes in the conclusion section and all other important areas that have been suggested in the revised manuscript. 

Thank you.

---

## [Decision Letter · Decision Letter 1]

24 Aug 2023

PONE-D-22-20978R1Title-Household Food Security access and Dietary Diversity amidst COVID-19 Pandemic in rural Nepal; an evidence from rapid assessmentPLOS ONE

Dear Dr. K.C.,

Thank you for submitting your manuscript to PLOS ONE. After careful consideration, we feel that it has merit but does not fully meet PLOS ONE’s publication criteria as it currently stands. Therefore, we invite you to submit a revised version of the manuscript that addresses the points raised during the review process.

We look forward to receiving your revised manuscript.

Kind regards,

George Vousden

Staff Editor

PLOS ONE

Journal Requirements:

Reviewers' comments:

Reviewer's Responses to Questions

**Comments to the Author**

1. If the authors have adequately addressed your comments raised in a previous round of review and you feel that this manuscript is now acceptable for publication, you may indicate that here to bypass the “Comments to the Author” section, enter your conflict of interest statement in the “Confidential to Editor” section, and submit your "Accept" recommendation.

Reviewer #1: All comments have been addressed

Reviewer #2: All comments have been addressed

2. Is the manuscript technically sound, and do the data support the conclusions?

Reviewer #1: Yes

Reviewer #2: Yes

3. Has the statistical analysis been performed appropriately and rigorously? 

Reviewer #1: Yes

Reviewer #2: Yes

4. Have the authors made all data underlying the findings in their manuscript fully available?

Reviewer #1: Yes

Reviewer #2: Yes

5. Is the manuscript presented in an intelligible fashion and written in standard English?

Reviewer #1: No

Reviewer #2: Yes

6. Review Comments to the Author

Reviewer #1: Commend the authors for going through a full revision of the manuscript. This paper's focus remains important and valuable. My main concern that remains is the use of causal language and language use to depict results which is not always consistent and thus confusing for the reader.

Assuming revised manuscript starts from page 43.

Page 43, Line 21: Please say 'COVID-19 pandemic' for specficity.

Line 34: Think you mean to say Household Dietary Diversity Score (HDDS)...score is missing.

Lines 34-36: Tenses used are not consistent - "whereas age and education are the predictors of HDDS", all other results are presented in past tense which makes sense. Also, when we say some is a predictor, there need some indication of directionality. Isn't the intent to say that age, ethnicity, educ, occupation are predictors of household food insecurity?

Lines 105-107: Here and in the abstract to there is still alot of use of causal language such as 'effects'. This is a cross-sectional and observational study. Any the analytic methods leveraged are not able to discuss any form of causal inference. Would temper language to just say that this is study examines the influence of the COVID-19 pandemic on household food security and household dietary diversity among rural communities in Nepal. Lines 109-113 again speak of impact.

Results section: The reporting of the results move back/ forth between discussing households as the unit of measurement to individual to families. Would encourage authors to provide some directionality when discussing associations or just state the ORs (lines 228-232).

Lines 238 + para that follows - Odds ratio do no reflect risk. The reflect odds. This section weaves in and our of reporting risk/odds.

Lines 289 - HFIAS is an indicator for food insecurity and HDDS is for dietary diversity, would simplify language to say thus say 'Despite the concerning high levels of food insecurity, household dietary diversity (HDDS>4) was low/medium for 63% of households in the study population and high for 36.8% of households.

Would recommend strengthening the strength/limitations section.

Reviewer #2: (No Response)

7. PLOS authors have the option to publish the peer review history of their article (what does this mean?). If published, this will include your full peer review and any attached files.

Reviewer #1: No

Reviewer #2: No

---

## [Author Response · Author response to Decision Letter 1]

3 Oct 2023

Journals requirements

Author’s response: We have checked all the included reference article used in this manuscript from the reference list. We couldn’t find a retracted article in our reference list and were published online in health journals. However, we hereby want to notify PLOS ONE team that we have updated the whole reference section in our updated manuscript. To meet the journals requirement we have made no significant changes has been made up to reference number 18, however from reference number 19-43 the references have been corrected, and updated which has been accompanied in our manuscript with track change and revised manuscript.

Pont by point response to #Reviewer 1

Reviewer #1: Page 43, Line 21: Please say 'COVID-19' pandemic for specificity.

Author’s response: Thank you for your suggestion. We have made the suggested changes.

Reviewer #1: Line 34: Think you mean to say Household Dietary Diversity Score (HDDS)...score is missing.

Author’s response: Thank you for your suggestion. We have made changes as per the reviewer’s suggestion.

Reviewer #1: Lines 34-36: Tenses used are not consistent - "whereas age and education are the predictors of HDDS", all other results are presented in past tense which makes sense. Also, when we say some is a predictor, there need some indication of directionality. Isn't the intent to say that age, ethnicity, educ, occupation are predictors of household food insecurity?

Author’s response: Thank you for your critical observation. We must have overlooked this during our initial revision. Tenses in the results section of abstract were adjusted as per the reviewer’s suggestion. Yes, we intend to present age, ethnicity, education, and occupation as the predictors of household food insecurity access. Amendment has been done in our revised manuscript following the suggestions of the reviewer. 

Reviewer #1: Lines 105-107: Here and in the abstract to there is still a lot of use of causal language such as 'effects'. This is a cross-sectional and observational study. Any the analytic methods leveraged are not able to discuss any form of causal inference. Would temper language to just say that this is study examines the influence of the COVID-19 pandemic on household food security and household dietary diversity among rural communities in Nepal. Lines 109-113 again speak of impact.

Author’s response: Thank you for your constructive feedbacks. We have adjusted the use of language such as “effects” and “impacts” as per the reviewer’s suggestion. 

Reviewer #1: Results section: The reporting of the results move back/ forth between discussing households as the unit of measurement to individual to families. Would encourage authors to provide some directionality when discussing associations or just state the ORs (lines 228-232).

Lines 238 + para that follows - Odds ratio do not reflect risk. The reflect odds. This section weaves in and out of reporting risk/odds.

Author’s response: Thank you for your comments and suggestions. As this study measures household food insecurity and dietary diversity among the households and the presented results are based on household as well as the participants. Some of our results needed to be presented based on the households characteristics whereas some needed to be presented based on the participants of the study. Hence, we have adjusted reporting of the results focusing on the households and individual participants interviewed wherever needed. We intended to present the socio-economic characteristics of our study participants in the line 228-232 based on the chi-square results obtained, their further association and ORs are discussed in the paragraph that follows whereby results of our bivariate and multi-variate analysis have been presented (line 239-248 of our tack change manuscript). We have adjusted our result section as per the reviewer’s suggestion and presented with reporting of odds wherever reviewer have pointed the need of reflecting odds. 

Reviewer #1: Lines 289 - HFIAS is an indicator for food insecurity and HDDS is for dietary diversity, would simplify language to say thus say 'Despite the concerning high levels of food insecurity, household dietary diversity (HDDS>4) was low/medium for 63% of households in the study population and high for 36.8% of households.

Author’s response: Thank you for your suggestion and directionality. We have adopted and modified line 289 exactly as per the reviewer’s suggestion.

Reviewer #1: Would recommend strengthening the strength/limitations section.

Author’s response: Thank you for your constructive comments. A line has been added to strengthen our strength/limitation section as per the reviewer’s suggestion.

---

## [Editor Report · Decision Letter 2]

16 Oct 2023

Household food security access and dietary diversity amidst COVID-19 pandemic in rural Nepal; an evidence from rapid assessment

PONE-D-22-20978R2

Dear Dr. K.C.,

We’re pleased to inform you that your manuscript has been judged scientifically suitable for publication and will be formally accepted for publication once it meets all outstanding technical requirements.

Kind regards,

George N Chidimbah Munthali

Academic Editor

PLOS ONE
---

## [Editor Report · Acceptance letter]

23 Oct 2023

PONE-D-22-20978R2 

Household food security access and dietary diversity amidst COVID-19 pandemic in rural Nepal; an evidence from rapid assessment 

Dear Dr. K.C.:

I'm pleased to inform you that your manuscript has been deemed suitable for publication in PLOS ONE. Congratulations! Your manuscript is now with our production department. 

Kind regards, 

on behalf of

Mr George N Chidimbah Munthali 

Academic Editor

PLOS ONE